# Symmetry mismatch-driven perpendicular magnetic anisotropy for perovskite/brownmillerite heterostructures

Jing Zhang[1,2], Zhicheng Zhong[3], Xiangxiang Guan[1,2], Xi Shen[1,2], Jine Zhang[1,2], Furong Han[1,2], Hui Zhang[1,2], Hongrui Zhang[1,2], Xi Yan[1,2], Qinghua Zhang[1,2], Lin Gu[1,2], Fengxia Hu[1,2], Richeng Yu[1,2], Baogen Shen[1,2] & Jirong Sun[1,2]

Grouping different transition metal oxides together by interface engineering is an important route toward emergent phenomenon. While most of the previous works focused on the interface effects in perovskite/perovskite heterostructures, here we reported on a symmetry mismatch-driven spin reorientation toward perpendicular magnetic anisotropy in perovskite/brownmillerite heterostructures, which is scarcely seen in tensile perovskite/perovskite heterostructures. We show that alternately stacking perovskite $La_{2/3}Sr_{1/3}MnO_3$ and brownmillerite $LaCoO_{2.5}$ causes a strong interface reconstruction due to symmetry discontinuity at interface: neighboring $MnO_6$ octahedra and $CoO_4$ tetrahedra at the perovskite/brownmillerite interface cooperatively relax in a manner that is unavailable for perovskite/perovskite interface, leading to distinct orbital reconstructions and thus the perpendicular magnetic anisotropy. Moreover, the perpendicular magnetic anisotropy is robust, with an anisotropy constant two orders of magnitude greater than the in-plane anisotropy of the perovskite/perovskite interface. The present work demonstrates the great potential of symmetry engineering in designing artificial materials on demand.

[1] Beijing National Laboratory for Condensed Matter and Institute of Physics, Chinese Academy of Sciences, Beijing 100190, People's Republic of China. [2] School of Physical Sciences, University of Chinese Academy of Sciences, Beijing 100049, People's Republic of China. [3] Key Laboratory of Magnetic Materials and Devices, Ningbo Institute of Materials Technology and Engineering, Chinese Academy of Sciences, Ningbo, Zhejiang 315201, People's Republic of China. These authors contributed equally: Jing Zhang, Zhicheng Zhong  Correspondence and requests for materials should be addressed to R.Y. (email: rcyu@iphy.ac.cn) or to J.S. (email: jrsun@iphy.ac.cn)

Transition metal oxides (TMOs) are known for their exotic properties stemming from strongly competitive mechanisms, such as ferromagnetic (FM) double exchange versus antiferromagnetic (AFM) superexchange, charge ordering vs. electron delocalization, electronic homogeneity vs. phase separation[1–4]. Grouping different TMOs into a heterostructure will break the delicate balance between different interactions, resulting in unforeseen effects[4,5]. To get an atomic level control of interface quality, usually the TMOs with similar crystal symmetry are chosen to compose artificial structures via interface engineering. Due to the excellent match in crystal symmetry and atomic configurations, the perovskite/perovskite (P/P) heterostructures have been intensively studied[6–18]. As revealed by Gibert et al.[18], the charge transfer from $LaMnO_3$ to $LaNiO_3$ makes the originally paramagnetic $LaNiO_3$ interfacial layer FM, yielding a magnetic pining to neighboring $LaMnO_3$. As shown by Liao et al.[19], transferring octahedron rotation from $NdGaO_3$ to $La_{2/3}Sr_{1/3}MnO_3$ (LSMO) caused an in-plane switching of the easy axis of LSMO by an angle of 90°; titling octahedron has modified the hopping rate of the $e_g$ electrons of Mn ions along the $a$- and $b$-axes, thus the magnetic anisotropy. As reported by Kan et al.[20], introducing a $Ca_{0.5}Sr_{0.5}TiO_3$ buffer layer to tune the network of the $RuO_6$ octahedra leads to a lateral 45° rotation of the easy axis of $SrRuO_3$.

Although hetero-phases have attracted increasing attention[21], so far most works on interface engineering focused on the P/P type heterostructure[9–20], a special heterostructure. In fact, there are many oxides that have similar structural framework and topotactic relationship as perovskite oxide but different atomic/electronic configurations. A typical example is the brownmillerite oxide with alternately stacked oxygen octahedra and tetrahedra[22–30]. Brownmillerite oxides have atomically ordered one-dimensional vacancy channel, showing advantages as ionic conductors, oxygen separation membranes, and catalyzers. In particular, some of them own the AFM order, providing a space for spin engineering with FM oxides. Because of the different symmetries of the two constituents, the perovskite/brownmillerite (P/B) heterostructure is expected to undergo considerable interface reconstruction, exhibiting various physical behaviors. Unfortunately, works on the effects of interlayer coupling in this kind of heterostructures are scarce, although several heterostructures have been fabricated[27, 31], probably due to the difficulty to obtain high quality P/B interfaces. Here we show an atomic level-controlled fabrication of the P/B type LSMO/$LaCoO_{2.5}$ (LCO) interfaces. The present work demonstrated the great potential of symmetry engineering in the exploration for emergent phenomena in magnetic complex oxides. Its application to, as an example, the LSMO/LCO multilayers has produced a strong perpendicular magnetic anisotropy (PMA), which has been in hot pursuit of spintronics, of the LSMO layer, which is otherwise of easy plane; the maximal perpendicular energy is $\sim 1.3\,J\,cm^{-3}$, which is more than one order of magnitude higher than that achieved via the conventional approaches such as magnetoelastic coupling (from 0.01 to $0.1\,J\,cm^{-3}$)[32–35] and magnetocrystalline anisotropy ($\sim 0.018\,J\,cm^{-3}$)[32]. This large PMA stems from the symmetry mismatch of the $MnO_6$ and $CoO_4$ layers at the LSMO/LCO interface, which results in cooperative distortions of the interfacial oxygen polyhedra as evidenced by high-resolution lattice structure analysis and density functional theory (DFT) calculations. Moreover, the symmetry break at the P/B interface is also expected to bring about emergent phenomena associated with directional electronic/ionic transport, two-dimensional electric polarization, and two-dimensional magnetism, etc., thus opening a promising avenue for the exploration for new concepts of physics and materials.

## Results

**Structure characterization of LSMO/LCO interfaces.** Heterostructures including LCO/LSMO/LCO trilayers and [LSMO/LCO]$_5$ superlattices (SLs) were grown on $TiO_2$-terminated (001)-$SrTiO_3$ (STO) single crystal substrates via pulsed laser deposition following the procedures described in the Methods section. Figure 1a presents the typical high-angle annular dark-field (HAADF) image of the cross-section of the LCO(5 nm)/LSMO(5 nm)/LCO(5 nm) trilayers, recorded along the [110] zone by scanning transmission electron microscope (STEM). Here, the brighter and fainter spots correspond to the La/Sr and Mn/Co atomic columns, respectively. Notably, parallel dark stripes appear every other column in the LCO layers. The La–La spacing is $\sim 4.3$ Å across the dark stripe and $\sim 3.5$ Å elsewhere in LCO, whereas the La/Sr-La/Sr spacing is $\sim 3.84$ Å in LSMO (Supplementary Figure 1). This kind of lattice image is reminiscent of that of brownmillerite oxides[22–26], which own a modulation structure with a periodicity of $4a_0$ along $c$ axis, where $a_0$ is the lattice constant of the perovskite unit cell. Indeed, the typical feature of the brownmillerite structure is identified from the HAADF image. As shown in Fig. 1a, the Co sites in dark stripes form paired structure, exhibiting breath mode lattice distortion with the intra- and inter-pair distances of $\sim 2.1$ Å and $\sim 3.4$ Å, respectively. A further finding is the staggered arrangement of the Co–Co pair in neighboring dark stripes (marked by green dots). These features are clearly seen in the DFT-calculated crystal structure of LCO (inset images in Fig. 1a, c and Supplementary Figure 2), confirming the brownmillerite character of the LCO layer. As an indication of the $LaCaO_{2.5}$ phase, shake up satellite peaks of the $Co^{2+}$ ions are also detected by X-ray photoelectron microscopy (XPS) for the LSMO/LCO multilayers (Supplementary Figure 1c)[36]. Different from LCO, the LSMO layer owns the typical perovskite structure.

A further important finding is that the dark stripe favors the position near LSMO. In other words, the interface in our heterostructures is of P/B type: $MnO_6$ octahedra prefer to connect to $CoO_4$ tetrahedra rather than $CoO_6$ octahedra. To confirm this conclusion, in Fig. 1b we show a HAADF image and the corresponding electron energy loss spectroscopy (EELS) spectrum images of the Mn-$L_{2,3}$ and Co-$L_{2,3}$ edges. The octahedra layer just below the LCO/LSMO interface is indeed composed of $MnO_6$, although there are slight interlayer diffusions. This result is consistent with the observation of Meyer et al.[27] who found that the first layer on the $TiO_6$ octahedron layer is composed of $CoO_4$ at the $SrCoO_{2.5}$/STO interface. Here we would like to emphasize that the LCO phase is unstable. It could be induced by interlayer interaction thus exist mainly in the proximity region of the LSMO/LCO interface.

It will be interesting to see how the $CoO_4$ tetrahedra and the $MnO_6$ octahedra accommodate each other at the LCO/LSMO interface. As shown in Supplementary Figure 3, just below the interface is a $MnO_6$ layer; each $MnO_6$ links to four neighboring $MnO_6$ octahedra through four corner oxygen atoms. On the other side of the interface, $CoO_4$ tetrahedra form another layer in parallel to that of $MnO_6$. However, each $CoO_4$ is only connected to two neighbors, forming $CoO_4$ chains without inter-chain connection. The distance is short between two Co atoms in this chain and long between chains, exhibiting a breath mode lattice distortion in the $CoO_4$ network as demonstrated by the HAADF image in Fig. 1a (green dots).

Due to symmetry mismatch, the lattice distortion at interface is expected to be strong and yet distinct. Figure 1c is an annular bright-field image taken along the [100] zone. The most remarkable finding is the misalignment of interfacial atoms: oxygen displays a visible upward shift ($\sim 0.6$ Å) rather than in line with its neighboring La/Sr atoms (marked by a red triangle). A

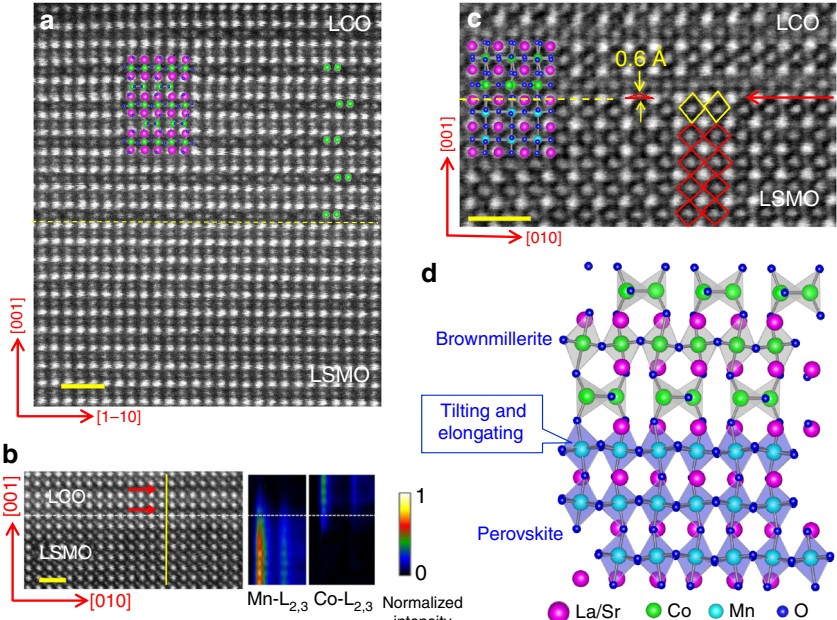

**Fig. 1** Lattice images of the LCO/LSMO/LCO trilayers. **a** High-angle annular dark-field (HAADF) image of the cross section of the LCO(5 nm)/LSMO(5 nm)/LCO(5 nm) heterostructure, recorded along the [110] zone. Brighter and fainter spots correspond to La/Sr and Mn/Co atomic columns, respectively. The lattice images with parallel dark stripes are LCO. The inset plot shows DFT calculated crystal structure of $LaCoO_{2.5}$. Its agreement with experimental observation indicates that the LCO layer is brownmillerite structured. The yellow dashed line marks the LSMO–LCO interface. **b** HAADF image and the corresponding EELS spectrum images of the $Mn-L_{2,3}$ and $Co-L_{2,3}$ edges, recorded along the yellow line. The interface between the LCO and LSMO layers is marked by a dashed line. Red arrows denote dark stripes. It clearly shows that the Mn–O monolayer locates just below the interface. **c** Annular bright-field (ABF) image of the cross section of the heterostructure, recorded along the [100] zone. The red triangle marks the misalignment of the interfacial La/Sr and O atoms. The yellow and red rhombuses denote the oxygen octahedra in interfacial and interior regions of LSMO, respectively. Inset plot: calculated crystal structure of $LaCoO_{2.5}$. O-La/Sr misalignment can be clearly seen at interface. The red arrow marks the LSMO-LCO interface. **d** A sketch of the brownmillerite structure and the LCO/LSMO interface, obtained by the DFT calculations. Scale bar, 1 nm

consequence of this is the elongation of the interfacial $MnO_6$ octahedra along $z$ axis. A quantitative analysis shows that the vertical size of the oxygen octahedra (yellow rhombuses) is ~ 4.2 Å at interface layer and ~ 3.8 Å elsewhere (red rhombuses). These experimental results are well reproduced by DFT calculations (inset image in Fig. 1c and the crystal structure in Fig. 1d). Notably, a $CoO_4$ in one layer has an exclusive corresponding $MnO_6$ in adjacent layer, connecting the latter through the apical oxygen atom (Supplementary Figure 2). As there is unfilled space in the $CoO_4$ layer, the neighboring $MnO_6$ will adjust its orientation and shape to minimize elastic energy. A result of this is the tiling around [110] axis and an accompanied elongation of the $MnO_6$ (Fig. 1d). In general, the elongation, in particular the octahedron tilting, will cause a chain reaction, reversely tilting the next $MnO_6$ layer. This means that interface effect is not limited to interfacial layer, extending to considerably distant layers though it will decay with the distance from interface. Obviously, at P/B interface the $MnO_6$ and $CoO_4$ polyhedra cooperatively relax in a distinct manner.

To further characterize the crystal quality and strained state of each constituent layer by X-ray diffraction (XRD), we prepared a series of LSMO/LCO SLs with a number of repetitions of five. Figure 2a shows the typical surface morphology of the typical SLs of LSMO(4 nm)/LCO(3 nm). The film surface is rather smooth, with the peak-to-valley height below 0.6 nm and the root-mean-square roughness of ~ 0.2 nm. Figure 2b presents the XRD spectra of two typical SLs with the LCO layer thickness of 3 nm and 8 nm (LSMO = 4 nm), respectively. Besides the main (002) reflection, satellite peaks (marked by numbers) and interference peaks (marked by triangles) can be clearly seen, confirming the high sample quality and the formation of SLs. Similar XRD spectra with

slightly different details were obtained for other SLs samples (Supplementary Figure 4). Figure 2c shows the reciprocal space mapping of the $(\bar{1}03)$ reflection for [LSMO(4 nm)/LCO(8 nm)]$_5$. The most remarkable feature is the vertical alignment for the reflections of the SLs and the substrate, i.e., the SLs share the in-plane lattice constant of 3.905 Å with substrate. Similar conclusions are applicable to other SLs. Based on the XRD spectra, we obtained the lattice parameters of the SLs. Figure 2d is the out-of-plane lattice constant of the SLs ($c_{SL}$) as a function of the layer thickness of LCO ($t_{LCO}$). Fitting the $c_{SL}$-$t_{LCO}$ relation to a simple equation of $c_{SL} = 4c_{LSMO}/(4 + t_{LCO}) + t_{LCO}c_{LCO}/(4 + t_{LCO})$ gives rise to the out-of-plane lattice parameters of $c_{LSMO} = 3.855$ Å and $c_{LCO} = 3.780$ Å. Both $c_{LSMO}$ and $c_{LCO}$ are obviously smaller than the in-plane lattice constant (3.905 Å), i.e., the LSMO and LCO unit cells are tensely strained (inset sketch in Fig. 2d).

**Magnetic anisotropy with interlayer symmetry mismatch.** The above results unambiguously show that the interface in our heterostructures is P/B type, totally different from the P/P interface. This is expected to have an impact on physical properties of the heterostructure. Indeed, we observed a symmetry mismatch-driven spin reorientation, which leads to a strong PMA that is not seen in the P/P heterostructure of 3d TMOs. To highlight the distinct effect of interlayer coupling, we performed a comparison investigation of the bare LSMO film and the LSMO/LCO heterostructure. Figure 3a, b illustrate the magnetic moments ($M$) of the bare LSMO film and the [LSMO(4 nm)/LCO(3 nm)]$_5$ SLs, respectively, as functions of temperature ($T$) (refer to Supplementary Figure 5 for the structural and magnetic properties of other bare LSMO and LCO films). The familiar $M$–$T$ dependence

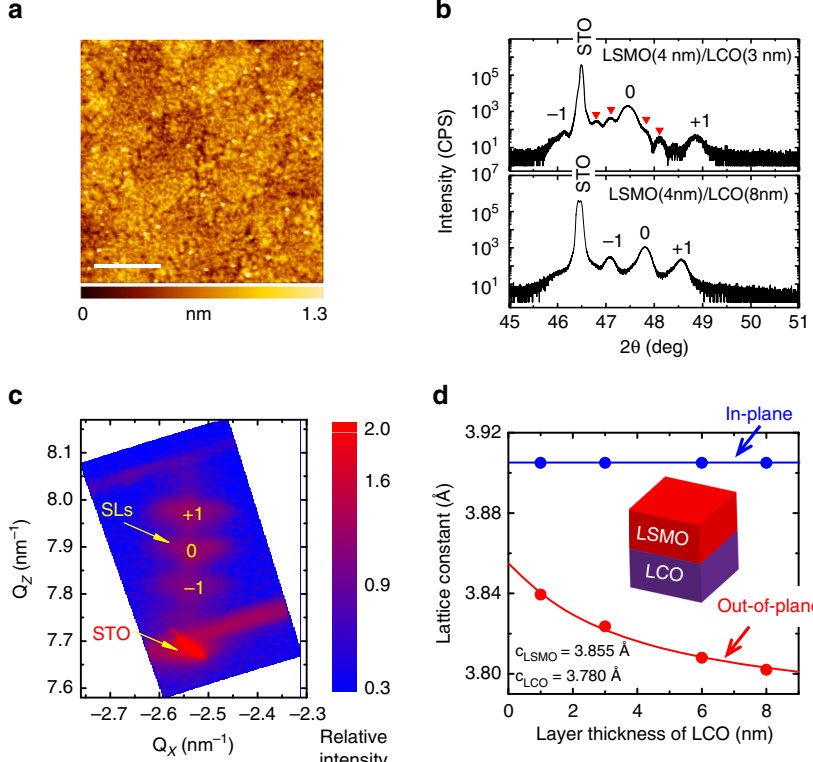

**Fig. 2** Structural characterizations of the LSMO/LCO SLs. **a** Surface morphology of the LSMO(4 nm)/LCO(3 nm) SLs. The film is flat with a root-mean-square roughness of 0.2 nm. Scale bar, 0.5 μm. **b** X-ray diffraction patterns of two typical SLs. Satellite peaks corresponding to superstructure (marked by numbers) and interferences due to finite film thickness (marked by red triangles) can be clearly seen. **c** Reciprocal space mapping of the (− 103) reflection of LSMO(4 nm)/LCO(8 nm). The vertical alignment for the reflections indicates the same in-plane lattice constant of the SLs as substrate. **d** Lattice parameters of the LSMO(4 nm)/LCO($t_{LCO}$) superlattices as functions of the LCO layer thickness. The deduced $c_{LCO}$ is smaller than the average value of the two A–A distances of the LCO determined by the STEM analysis. Possibly, the brownmillerite phase in the SLs prefers to form in the proximity region of the interface and the LCO layers are not totally of the brownmillerite structure

is seen for the LSMO film when applying magnetic field ($H$) along film plane (Fig. 3a, the $M$–$T$ curve marked by IP): the magnetic moment takes the maximal value at low temperatures, decreases smoothly upon warming when $T$ is well below 320 K, and rapidly close to 320 K, the Curie temperature of LSMO. In contrast, perpendicular fields induce substantially low magnetic moments unless it is high enough. Clearly, the easy axis of the bare LSMO film is in film plane.

In sharp contrast to the in-plane magnetic behavior of the LSMO film, an anomalous PMA is indicated by the $M$–$T$ relations of the SLs. Take the data recorded with a field of 0.2 T as an example. Unexpectedly, the in-plane magnetic moment is low at low temperatures and smoothly increases upon warming until a local maximum is reached. After that, a rapid decrease appears. On the contrary, the out-of-plane magnetic moment is substantially high, larger by a factor of 2.6 than the in-plane one at 10 K. Increase in $T$ causes a slow decrease of it but its superiority remains obvious up to ~ 200 K. Apparently, the spins of the P/B heterostructure prefer to align perpendicularly, rather than laterally like the bare LSMO film (Fig. 3a), which is the most remarkable observation of the present work. At 220 K, the lateral moment equals to the perpendicular one, indicating that the directional preference of the magnetic moment is not strong. The increase-to-decrease crossover at ~ 240 K in the $M$–$T$ curve (Fig. 3b) is an indication of the in-plane to out-of-plane spin re-orientation. Above ~ 240 K, the heterostructure returns to the normal state of the LSMO film.

To get a quantitative description of magnetic anisotropy, anisotropy constant ($K_{tot}$) is calculated. Depicting $M$ as a function of $H$ at fixed temperatures, we obtained a series of $M$–$H$ curves based on the data in Fig. 3b (see Supplementary Figure 6 for directly measured $M$–$H$ loops). Figure 3c present two typical $M$-$H$ relations at 10 K, obtained with perpendicular and parallel fields, respectively. As shown, a saturation state is established in a perpendicular field of 0.2 T, whereas it is not reached in a parallel field close to 3 T. The energy required to laterally orientate the magnetic moments can be calculated from the area encircled by the two $M$–$H$ curves in Fig. 3c. A direct calculation gives the anisotropy constant of $K_{tot} = 5.4 \times 10^6$ erg cm$^{-3}$. The positive sign of $K_{tot}$ implies PMA. Surprisingly, we obtained the PMA in tensile LSMO layer by sandwiching it between two LCO layers. For the SLs, the magnetic contributions could mainly come from LSMO, as no obvious magnetic transition is detected at the $T_C$ of LCO (~ 75 K)[37]. Following the similar procedure, we obtained the $K_{tot}$ at high temperatures. Figure 3d shows the $K_{tot}$ as a function of temperature. It is maximal at low temperature, slowly decreases upon warming, and changes its sign at ~ 220 K. Above 220 K, PMA is no longer supported and the easy plane state prevails. For comparison, the $K_{tot}$ of the plain LSMO film is also shown in Fig. 3d. Its negative sign implies the expected lateral spin orientation. In fact, the interfacial $K_{tot}$ of the SLs could be even larger than that shown here, as it has to overcome the intrinsic lateral spin preference of the LSMO itself. With this consideration, the actual $K_{tot}$ of the SLs could be $K_{tot}$(SLs) – $K_{tot}$(LSMO), as

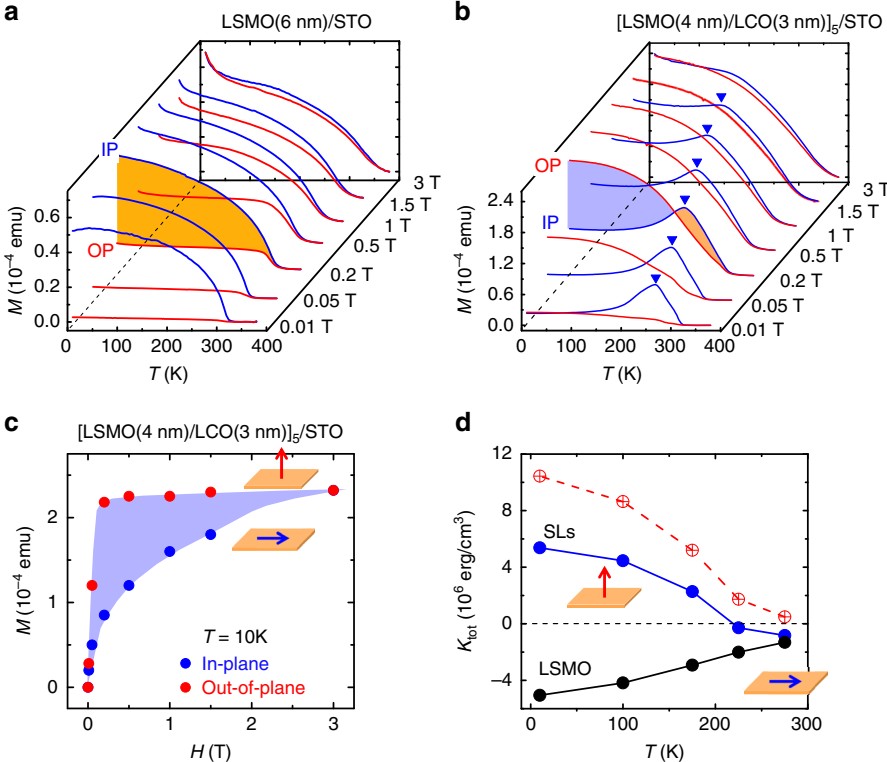

**Fig. 3** Magnetic behaviors of the LSMO film and the LSMO/LCO SLs. **a,b** Thermomagnetic curves of the LSMO film (6 nm) and the LSMO(4 nm)/LCO(3 nm) SLs, respectively. The data were acquired in field-cooling mode with in-plane (IP) or out-of-plane (OP) applied fields. Purple and orange areas highlight the difference of the magnetic moments along two measuring directions. Two $M$–$T$ curves recorded along the in-plane and the out-of-plane directions, respectively, sometimes cross one another, as the spin re-orientation has changed relative variations of the magnetic moment with temperature along these two directions. Blue triangles mark the temperature for spin reorientation. **c** Magnetic moment as a function of applied fields, extracted from the data in **b** at $T = 10$ K. Shaded area corresponds to the energy required to orientate magnetic moment towards film plane. Here, the data at 10 K were presented simply because that our measuring system is most easily thermally stabilized at this temperature. At low temperatures well below the Curie temperature, the magnetic properties of the multilayers are nearly temperature independent. **d** Anisotropy constant of the LSMO(4 nm)/LCO(3 nm) SLs (blue curve) and the plain LSMO film (black curve). The expected anisotropy constant of the SLs is presented by a dashed line

shown by the dashed line in Fig. 3e. It reaches a value as high as $1.04 \times 10^7$ erg cm$^{-3}$ at 10 K.

Further investigations show that the anomalous PMA arises from interface effects. Varying the layer thickness of LSMO from $t_{LSMO} = 3$ nm to 15 nm while fixing $t_{LCO}$ to 6 nm, in Fig. 4a we show the $M$–$T$ curves collected with an applied field of 0.05 T for the LCO(6 nm)/LSMO($t_{LSMO}$)/LCO(6 nm) trilayers, which own the main characters of the SLs but has a simple structure. The strongest PMA appears when $t_{LSMO} = 3$ nm: the out-of-plane moment overwhelms the in-plane one in the whole temperature range investigated. However, in-plane component develops with the increase of the LSMO thickness and finally overtakes the out-of-plane one when $t_{LSMO} = 15$ nm. To determine $K_{tot}$, $M$–$T$ curves are also measured in higher fields for all trilayers (Supplementary Figure 7) and the $K_{tot}$–$t_{LSMO}$ dependence is obtained. As shown by Fig. 4b, $K_{tot}$ exhibits a well linear dependence on $1/t_{LSMO}$. This is the fingerprint of interface-induced magnetic anisotropy[38], strongly suggesting that the PMA is stabilized by interfacial effect. According to Fig. 4b, the maximal apparent $K_{tot}$ is ~ $7.5 \times 10^6$ erg cm$^{-3}$. Accounting for the negative contribution to anisotropy constant of the LSMO film, the maximal $K_{tot}$ will be ~ 1.3 J cm$^{-3}$.

Notably, the PMA in our heterostructure persists significant up to the layer thickness of 10 nm (Fig. 4b) in contrast to 2 nm for metallic films composed of metals and/or alloys[38], and it not only is easily controlled but also will satisfy special requirements for thick PMA films.

To clarify the effect of LCO on interlayer coupling, we further investigated the magnetic behavior of the LCO($t_{LCO}$)/LSMO(6 nm)/LCO($t_{LCO}$) trilayers with varied LCO layer thickness. The most remarkable observation is the presence of a threshold LCO thickness for the PMA, i.e., the LCO–LSMO interlayer coupling takes effect only for thick enough LCO. As shown in Fig. 4c (refer to Supplementary Figure 8 for high field data), interlayer coupling is so weak when the LCO layer is 0.8 nm that the spins of LSMO persist to lie in film plane. It takes effect when LCO is 3 nm and a clear signature of spin re-orientation can be identified from the $M$–$T$ curves. The strongest effect occurs when $t_{LCO} = 6$ nm for this series of trilayers. The corresponding $K_{tot}$ is $4.5 \times 10^6$ erg cm$^{-3}$ and the PMA weakens again when $t_{LCO} > 6$ nm. This result shows that a too thin or a too thick LCO layer in the multilayers disfavors the PMA. Shown in Fig. 4d is the $K_{tot}$–$t_{LCO}$ dependence. The threshold thickness for interlayer coupling is ~ 1.5 nm.

We examined the EELS spectra of the multilayers and believed that the weak interlayer diffusion is unable to destroy the effect of LCO on LSMO. Possibly, the brownmillerite structure is not well established when LCO is 0.8 nm, as the $c$ axis lattice constant is ~ 1.5 nm for a brownmillerite unit cell. Therefore, the P/B interface is crucially important for the PMA.

As a supplement, we noted that in Fig. 4a, b the Curie temperature of the trilayers monotonically grows with $t_{LSMO}$ and keeps nearly constant as $t_{LCO}$ increases (Supplementary Figure 9). This is understandable since the Curie temperature of the trilayers is mainly determined by the LSMO layer.

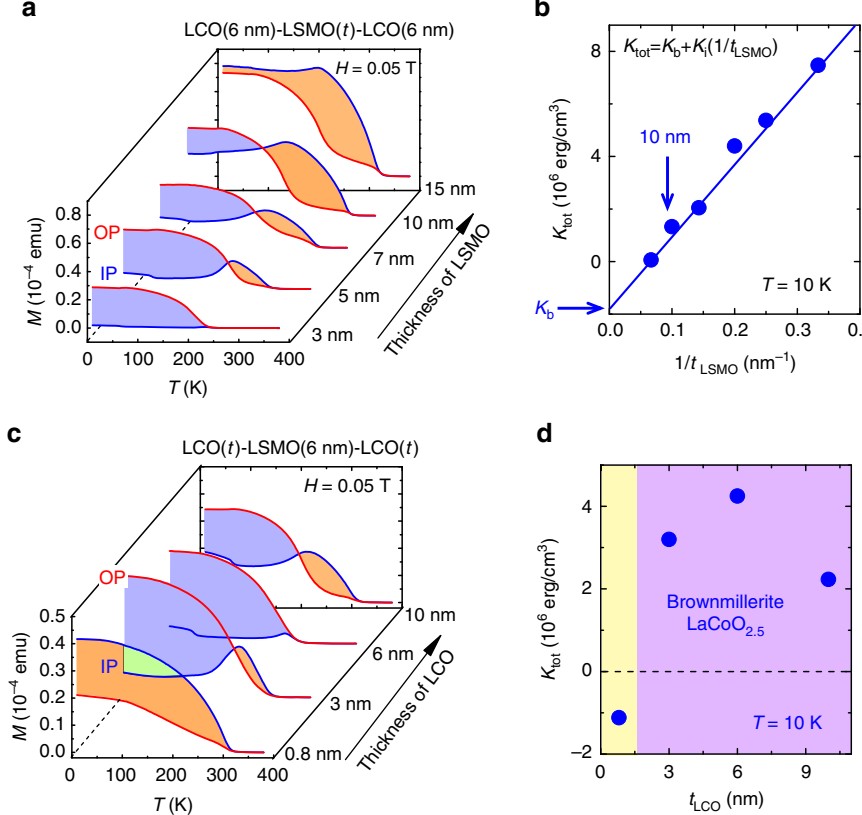

**Fig. 4** Effects of layer thickness on magnetic behaviors for LCO/LSMO/LCO trilayers. **a** Thermomagnetic curves of the LCO(6 nm)/LSMO($t_{LSMO}$)/LCO(6 nm) trilayers, collected in field-cooling mode with an in-plane (IP) and an out-of-plane (OP) applied field of 0.05 T, respectively; here, $t_{LSMO}$ takes a value between 3 and 15 nm. Shaded area corresponds to the energy required to orientate magnetic moment to hard axis. **b** Magnetic anisotropy energy as a function the layer thickness of LSMO. The linear $K_{tot} - 1/t_{LSMO}$ relation is a fingerprint of interfacial effect, where $K_{tot}$ is decomposed into $K_b$ and $K_i$, corresponding to the bulk and interface contributions, respectively. **c** The same as **a**, except for samples that are now LCO($t_{LCO}$)/LSMO(6 nm)/LCO($t_{LCO}$) with a $t_{LCO}$ ranging from 0.8 to 10 nm. **d** Magnetic anisotropy energy as a function of the layer thickness of LCO. Here, the data at 10 K were presented, simply because our measuring system is most easily thermally stabilized at this temperature

**Magnetic anisotropy without interlayer symmetry mismatch.** To confirm this unusual PMA effect, we further studied the magnetic behavior of P/P-typed multilayers. Figure 5a is the typical HAADF image of the LCO(6 nm)/LSMO(6 nm)/LCO(6 nm) trilayers fabricated under an oxygen pressure of $P_{O_2} = 50$ Pa. Remarkably, dark stripes and the corresponding breath mode lattice distortions for Co atoms completely vanish, indicating the absence of cooperative CoO_4 distortions. This means that the LCO/LSMO interface is sandwiched by a MnO_6 and a CoO_6 layer, i.e., it is of P/P type in nature. Figure 5b is the $M–T$ curves collected with an applied field of 0.05 T for the LCO/LSMO/LCO trilayers fabricated under different $P_{O_2}$s. It reveals an obvious weakening of the PMA as $P_{O_2}$ exceeds 40 Pa. The evolution of the anisotropy energy with oxygen pressure is shown in Fig. 5c. $K_{tot}$ is low when $P_{O_2}$ is 20 Pa due to the degeneration of the magnetism of LSMO when fabrication oxygen pressure is low, and maximizes at 30 Pa. However, further increase in $P_{O_2}$ causes a rapid decrease in $K_{tot}$, and the in-plane magnetic anisotropy prevails when $P_{O_2}$ is 50 Pa.

As stated above, the typical shake-up satellite peaks of the Co$^{2+}$ ions have been observed in the XPS spectrum of the trilayers with modulated lattice structures, which are the lateral evidence of the B-typed structure of the LCO layer. In contrast, no satellite peaks are observed for the LCO/LSMO/LCO trilayers prepared under 50 Pa (Supplementary Figure 1c). This is consistent with the result of STEM analysis, which suggests that the LCO layer in this sample is perovskite structured.

**DFT calculations.** As well established, shape anisotropy, magnetoelastic coupling, and magnetocrystalline anisotropy are the main sources for magnetic anisotropy. For our LSMO/LCO SLs and trilayers, the first two mechanisms will favor an in-plane spin orientation, hence cannot explain the observed PMA. Magnetocrystalline anisotropy is closely related to crystal field anisotropy and orbital anisotropy. As demonstrated by Fig. 1, the P/B-type interface in our heterostructures has resulted in substantial symmetry breaking and unique lattice distortions, and is therefore expected to cause unusual magnetocrystalline anisotropy. We performed the DFT calculations for the SLs composed of alternately stacked 3-uc-LSMO and 3-uc-LCO (3/3-LSMO/LCO SLs, see Supplementary Figure 10 for details), which contain P/B-type interfaces as shown in Fig. 6. Various magnetic structures including the FM, A-type AFM, and G-type AFM ones were investigated. Our calculation confirmed the FM ordering in the LSMO layer and found that the easy axis is in the normal direction of the film plane, with the magnetic anisotropy energy (MAE) of 0.15 meV/Mn ($\sim 4 \times 10^6$ erg cm$^{-3}$). To highlight the effect of the P–B interface, we also conducted the computation for the 1/1-LSMO/LCO SLs (Supplementary Figure 10) and obtained an MAE of 0.80 meV/Mn ($\sim 2.1$ J cm$^{-3}$), which is comparable to the experimentally obtained value (1.3 J cm$^{-3}$). As a reference, the magnetic anisotropy of the P/P-type SLs (1/1-LSMO/LaCoO_3) was also investigated and an easy axis in film plane is obtained (Supplementary Figure 10). Clearly, it is the P/B interface that causes the PMA.

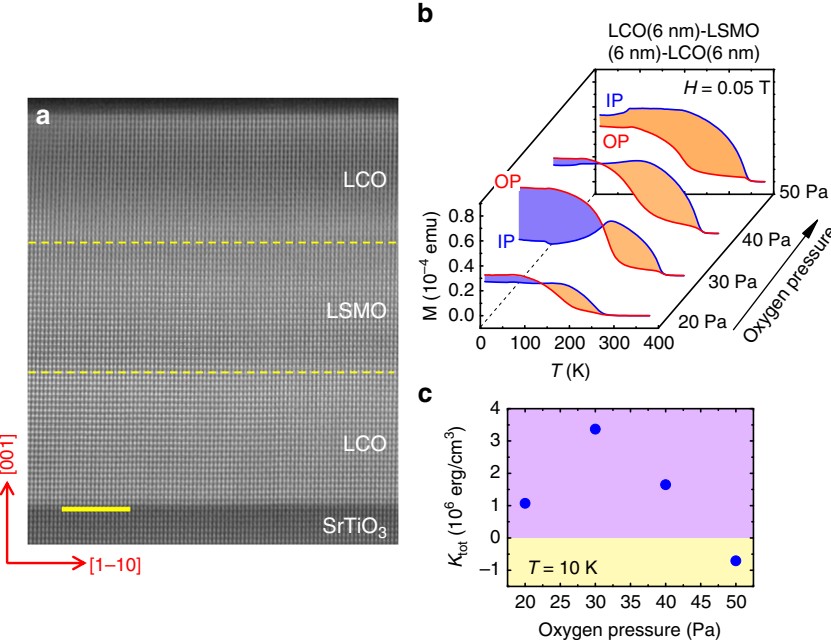

**Fig. 5** Effects of oxygen pressure on LCO/LSMO/LCO trilayers. **a** A high-angle annular dark-field (HAADF) image of the cross section of the LCO(6 nm)/LSMO(6 nm)/LCO(6 nm) trilayers fabricated under an oxygen pressure of 50 Pa, showing a P/P type interface as marked by yellow dashed lines. Scale bar, 5 nm. **b** Thermomagnetic curves for the LCO/LSMO/LCO trilayers prepared under different $P_{O_2}$s, collected in field-cooling mode with an in-plane (IP) and an out-of-plane (OP) applied field of 0.05 T, respectively. In-plane magnetic anisotropy is observed under the $P_{O_2}$ of 50 Pa. Here, the layer thickness is a nominal one. The actual thickness is slightly thicker than the nominal one according to the HAADF image. **c** Magnetic anisotropy energy as a function of oxygen pressure

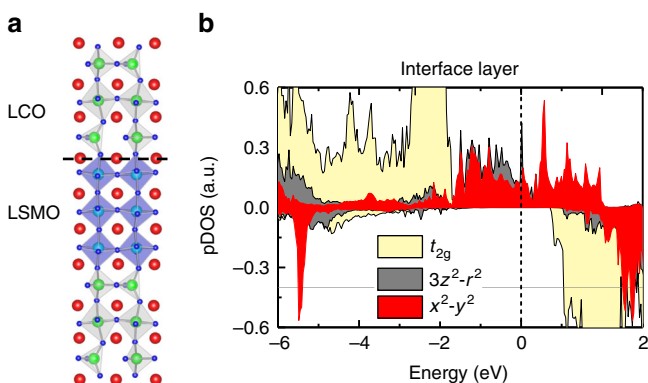

**Fig. 6** Results of density functional theory calculations. **a** DFT calculated 3/3-LSMO/LCO superlattices with two $CoO_4/MnO_6$ interfaces. **b** Projected density of Mn 3d states of the interfacial layer. $d_{3z^2-r^2}$ states are marked in gray, $d_{x^2-y^2}$ in red, and $t_{2g}$ in yellow

## Discussion

A further issue to be addressed is the physical origin of the PMA. According to the result of STEM analysis and theoretical calculations, the interfacial $MnO_6$ octahedra are elongated along the [001] axis and tilted around the [110] axis to accommodate the symmetry mismatch between the $CoO_4$ layer and the $MnO_6$ layers (Fig. 6a and Supplementary Figures 2 and 3). Meanwhile, charge transfer may take place between Mn and Co, building up an out-of-plane Co-O-Mn covalent bond[31] (charge transfer has been observed across the $(Y,Ca)Ba_2Cu_3O_7/La_{2/3}Ca_{1/3}MnO_3$ interface, resulting in an orbital reconstruction)[39]. All these will cause a preferred occupation of the $d_{3z^2-r^2}$ orbital, as confirmed by the DFT calculations (Fig. 6b). This effect will be especially strong

at low temperatures, which depress the $d_{3z^2-r^2}$ to $d_{x^2-y^2}$ excitation. In this case, the orbital momentum could be finite in the normal direction of the film plane[40]. According to the Bruno model[41,42], easy axis prefers to take the direction of the orbital momentum. This explains the occurrence of the PMA and why the PMA prefers to appear at low temperatures.

Through tilting oxygen octahedra at the P/P type interface, researchers have achieved a successful control of the lateral anisotropy for the typical perovskite heterostructures[19,20]. Due to the similarity of the oxygen polyhedra in two sides of the interface, however, the tuning effect is generally subtle: the typical anisotropy constant is in the order of $10^4 \, erg \, cm^{-3}$, as for the LSMO film on $NdGaO_3$ [19]. Compared with P/P heterostructures, P/B is a promising combination. The A or B sublattice of these two oxides ($ABO_3$ and $ABO_{2.5}$) well matches each other, which allows epitaxial film growth, whereas the oxygen polyhedra symmetry at interface is completely different. This produces a great and unique interfacial reconstruction (Fig. 1d and Supplementary Figures 2 and 3), thus a strong impact on physical behaviors.

As stated above, the PMA is important because of its irreplaceable role in high-density information storage, current-induced magnetic switching[43,44], and efficient spin light-emitting diode[45]. As an example, a natural extension of the LSMO/LCO SLs may be perpendicular magnetic tunnel junctions where the LSMO layers serve as electrodes with a perpendicular spin orientation and the LCO layer acts as a barrier layer that is highly insulating at low temperatures. For this kind of magnetic junction, it will be much easier for electrical current to reverse the spin direction of the LSMO layer, yielding electric field-controllable magnetoresistance (we have investigated the transport behavior of the SLs. With the decrease of temperature, the resistance of the SLs is well metallic below 300 K like bulk LSMO). An advantage of the present SLs over the typical metallic film for perpendicular magnetic tunnel junctions is that its PMA

has a large tolerance of film thickness, occurring in the thickness range from 3 to 10 nm, rather than being limited to ~ 2 nm[43]. Moreover, noting the unique characters of the brownmillerite oxides as ionic conductors, catalyzers, and oxygen separation membranes, amazing effects[29] in addition to PMA are expected for the P/B combinations. For example, the symmetry break at the P/B interface may bring about emergent phenomena associated with directional electronic/ionic transport, two-dimensional electric polarization and two-dimensional magnetism, etc., thus opening a promising avenue for the exploration for new concepts of physics and materials.

In summary, an atomic level controlled fabrication of the P/B-type LSMO/LCO heterostructure has been demonstrated and a symmetry mismatch-induced spin reorientation towards PMA has been observed even the heterostructure is in an in-plane tensile state. The easy axis can be deliberately tuned between parallel and perpendicular directions by altering interface effects. It is found that stacking perovskite and brownmillerite oxides alternately caused a strong and distinct interface reconstruction due to symmetry discontinuity at interface: neighboring $MnO_6$ octahedra and $CoO_4$ tetrahedra at the interface cooperatively relax in a manner that is unavailable for the P/P interface, resulting in unique orbital reconstructions, thus the PMA. The present work demonstrates the distinct effect arising from the interface between oxides of different symmetry. The principle proven here can be extended to other combinations of TMOs, substantially extending the space for the exploration of emergent phenomena.

## Methods

**Sample preparation**. LSMO/LCO heterostructures were grown on $TiO_2$-terminated (001)-STO single crystal substrates ($3 \times 5 \times 0.5$ mm$^3$), using the technique of pulsed laser ablation. The fluence of the laser pulse was 2 J cm$^{-2}$ and the repetition rate was 2 Hz (KrF Excimer laser, wavelength = 248 nm). The substrate temperature was maintained at 700 °C (for LSMO) or 635 °C (for LCO), and the oxygen pressure ($P_{O_2}$) was fixed to a constant value of 30 Pa. Here, a low growth temperature was adopted for the LCO layer to avoid recrystallization, which will yield a rough interface. After deposition, the samples were cooled to room temperature at a rate of 10 °C min$^{-1}$ in an oxygen atmosphere of 500 Pa. Film thickness was determined by the number of laser pulses, which has been carefully calibrated by the technique of small angle X-ray reflectivity and STEM.

Following this procedure, we fabricated three sets of samples. The first set samples are four LSMO/LCO SLs with a periodicity of 5. The layer thickness is 4 nm for LSMO and 1, 3, 6, and 8 nm for LCO. For all SLs, the bottom and top layers are LCO and LSMO, respectively. The second set samples were five LCO/LSMO/LCO trilayers, comprising a LSMO layer ranging from 3 to 15 nm sandwiched between two LCO layers of 6 nm. The third set samples were four LCO/LSMO/LCO trilayers, with a LSMO layer of 6 nm and a LCO layer between 0.8 and 10 nm. The latter two sets of samples were prepared to investigate the effect of layer thickness.

To investigate the effect of oxygen pressure, we fabricated the fourth set of LCO/LSMO/LCO trilayers following the same procedure as described above but for different oxygen pressures, which were set to 20, 30, 40, and 50 Pa, respectively. The layer thickness was set to 6 nm, the same for the LSMO and the LCO layer.

**Structural and magnetic measurements**. Surface morphology of the heterostructure was measured by atomic force microscope (SPI 3800N, Seiko). Crystal structure of the films was determined by a Bruker X-ray diffractometer equipped with thin film accessories (D8 Discover, Cu Kα radiation). Lattice images were recorded by a high-resolution STEM with double C$_S$ correctors (JEOL-ARM200F). Magnetic measurements were conducted by a Quantum Designed Vibrating Sample Magnetometer (VSM-SQUID) in the temperature interval from 5 to 300 K and the magnetic field range up to 7 T.

**DFT calculations**. The DFT calculations were performed with the Vienna ab initio simulation package code[46] using the generalized gradient approximation GGA-PBEsol functional[47] for electronic exchange and correlation. A kinetic energy cutoff of 500 eV was used and the Brillouin zone was sampled with a $9 \times 9 \times 3$ $k$-point grid in combination with a tetrahedron method. We performed the calculations including an on-site Coulomb repulsion term of $U = 2$ eV for both Mn and Co $d$ electrons. We investigated various spin configurations, including FM, A-type AFM, and G-type AFM. As our LSMO/LCO SLs and LCO/LSMO/LCO interfaces have the same in-plane lattice constant with STO, we simulated the strain effect by fixing the in plane lattice constant to 3.905 Å unless specifically stated. The out-of-plane lattice constant was taken from experimental values deduced from the XRD and STEM data. All the internal atomic positions are optimized in the presence of the FM order. Moreover, we performed the calculations including spin-orbit coupling to describe magnetocrystalline anisotropy, which is expected to have the dominant role for the observed PMA.

**Data availability**. The data that support the findings of this study are available from the corresponding author upon request.

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

## Acknowledgements

This work has been supported by the National Basic Research of China (Nos. 2016YFA0300701, 2017YFA0206300, 2017YFA0303602 and 2017YFA0303600), the National Natural Science Foundation of China (Nos. 11520101002, 51590880, and 11674378), and the Key Program of the Chinese Academy of Sciences.

## Author contributions

J.R.S. conceived and designed the experiments, interpreted, together with Z.C.Z., the experimental results, and prepared the manuscript. J.Z. conducted most of the experiments. J.E.Z., F.R.H., H.Z., and H.R.Z. performed some magnetic measurements. X.X.G., X.S., R.C.Y., Q.H.Z., and L.G. undertook the analysis of transmission electron microscope. Z.C.Z. carried out the DFT calculations. X.Y. and F.X.H. characterized the sample via atomic force microscope. B.G.S. oversaw the project. All authors commented on the manuscript.

## Additional information

**Competing interests:** The authors declare no competing interests.

