## [Peer Review File · Nature Communications]

Reviewers' Comments:

Reviewer #1:

Remarks to the Author:

The submitted paper by J. Zhang, et al. prepared heterostructures consisting of the $\text{La}_{2/3}\text{Sr}_{1/3}\text{MnO}_3$ perovskite and the $\text{LaCoO}_{2.5}$ brownmillerite. The authors analyzed the interface of the fabricated heterostructure and found the characteristic interface structure where the CoO_4 tetrahedra connect to the MnO_6 octahedra, (although a similar interface structure was already reported in a previous paper [Ref. 26]). The authors further made superlattices, measured the magnetic properties, and found perpendicular magnetic anisotropy in some of the fabricated superlattices with specific thicknesses of constituent layers. With the results the authors concluded that the modified interface of layers with different structure symmetries causes the change in magnetic anisotropy of the $\text{La}_{2/3}\text{Sr}_{1/3}\text{MnO}_3$ films.

The authors appear to have made the enough number of heterostructures including superlattices with different thicknesses and to have conducted the measurements properly. The observed experimental results including the perpendicular magnetic anisotropy are of particular interest. However, the reviewer is not convinced with the authors' discussion and conclusion.

The authors referred to Ref. 31, which reported the magnetic transition of LaCoO_3 at about 75 K, and concluded that the $\text{LaCoO}_{2.5}$ did not contribute the magnetic properties of their heterostructures. It is well known that LaCoO_{3-d} ($\text{LaCoO}_{2.5+d}$) show ferromagnetic transitions up to about 290 K in bulk (S. Balamurugan, et al., PRB 74, 172406, 2006) and 300 K in thin films (N. Ichikawa, et al., Dalton Trans. 41, 10507, 2007). Those previous results indicate that CoO_6 octahedra give ferromagnetic transitions at 290-300 K, which are very close to the magnetic transition temperature of $\text{La}_{2/3}\text{Sr}_{1/3}\text{MnO}_3$ in the present study.

Therefore, there still be a possibility that the observed perpendicular magnetic moments of the fabricated superlattices originated from the (partially) oxidized $\text{LaCoO}_{2.5}$ film layer, not the $\text{La}_{2/3}\text{Sr}_{1/3}\text{MnO}_3$ layer. Actually, the oxygen partial pressure during the deposition for $\text{LaCoO}_{2.5}$ in the present study (30 Pa) is quite high compared to that reported in a previous paper.

The presence of CoO_6 octahedra near the interface should be checked carefully. The observed elongated octahedra might be CoO_6 not MnO_6 . The authors are advised to conduct an atomic resolution EELS analysis in the STEM observation and to confirm the exact position of interface (not from the image) and the elongated octahedral is MnO_6 .

The above discussion is essential for clarifying the origin of the observed perpendicular magnetic moments. Without the confirmation the submitted paper should not be accepted for publication in Nature Communications.

An additional comment:

Figure 2d is difficult to understand, and the authors are requested to add more explanation. Which kinds of lattice parameters are plotted in blue and red in this figure? The extrapolated values (in red?) seem to correspond to the bulk lattice parameters for LSMO (at LCO layer thickness 0) and LCO (at ∞), but the results do not. The obtained $c(\text{LCO}) = 3.77 \text{ \AA}$ does not match either value of A-A distance in Figure 1b. Details explanations are needed.

Reviewer #2:

Remarks to the Author:

Report corresponding to the manuscript

"Symmetry mismatch-driven perpendicular magnetic anisotropy for perovskite/brownmillerite heterostructures"

By Jing Zhang et al.

In this manuscript, the authors show the control of the spin orientation from in-plane to perpendicular magnetic spin orientation in transition metal oxide heterostructures. This behavior is achieved through the growth of unusual heterostructures combining perovskite La(Sr)MnO₃ and brownmillerite LaCoO_{2.5} layers. The achievement of perpendicular magnetic anisotropy results from strong interfacial reconstructions occurring at the unconventional perovskite/brownmillerite interface. Transmission microscopy images evidence the interfacial reconstructions. Detailed SQUID analyses on a variety of superlattices with different layer-periodicities point to structural interfacial reconstructions, leading to modified orbital occupation, as the mechanism to achieve perpendicular magnetic anisotropy. DFT calculations are also included.

One of my main comments is why interfacial charge transfer is not considered in this system, whereas it is known to occur in other Mn/Cu interfaces such as YBCO/LCMO (i.e. Chakhalian et al., Science, 318 (2007) 1114-1117). Did the authors investigate it through X-ray Absorption Spectroscopy (XAS) or Electron Energy Loss Spectroscopy (EELS)? What about orbital polarization?

It would be useful for the reader to remind the bulk properties of the La(Sr)MnO₃ and LaCoO_{2.5}. What about the magnetic properties of the LaCoO_{2.5} films? In the manuscript it is simply mentioned that no signatures of LCO T_c are observed about the superlattices.

The authors should also comment on the transport properties of the superlattices investigated. In line 228, the authors mention that their heterostructures' properties contrast with those of metallic films. Are their La(Sr)MnO₃-based superlattices not metallic?

As indicated by the authors, brownmillerite structures are not as commonly investigated as perovskites. Therefore, I suggest to introduce sketches of the brownmillerite structure as well as the interface in the main figures of the paper.

Other details:

- Line 26: "relax" instead of "relaxed"
- Line 125: "...LSMO&LCO SIs with a periodicity of 5...": I guess it should be "number of repetitions" instead of "periodicity"
- Line 142: goal of sketch showing tensile strained is not achieved
- Fig. 3c: explicitly indicate that it corresponds to superlattices (not LSMO)
- Line 228: is it Fig. 4b-citation correct?

The results are interesting and novel, and I think the manuscript deserves publication in this journal.

Reviewer #3:

Remarks to the Author:

The manuscript by Zhang et al reports on magnetic properties of LSMO/LCO multilayers. The thickness dependence of anisotropy is evaluated in this study and is the primary focus here with relevance to spintronics. I have several comments for the authors:

- The symmetry mismatch leads to distinct interfacial structures, such phenomena has been discussed in literature for heterophases, e.g Wong et al, J Vac Sci Tech A, 32, 040801, 2014. It will be important for the authors to describe the structure, defects and how the symmetry mismatch is accommodated. This is not clear just looking at the TEM image in figure 1. What kinds of extended defects are formed for instance?

- There is no discussion on how inter-diffusion can affect the properties reported here. For instance, the authors claim there is no effect for the first few layers, and then the anisotropy effect

kicks in. It will be important to explain to the reader why this is the case. Is this related to strain, chemical in-homogeneity, unknown phase formation at interface etc.

- Thermo-magnetic plots are shown for various fields and T in the main and supplement. It will be useful for the authors to analyse this and present a summary. for instance, is the T_c changing with interfacial layer structure or number of layers? In some cases, I see the curves crossing one another, but this is not explained.

- It will be helpful for the authors to elaborate on the principal result. It is not clear to me why this result is particularly significant or some comparison to existing spintronic technologies or how the anisotropy constant compares to what is useful or impressive in this class of materials or others etc? Without this information properly discussed in the context of the introduction and claims, it is not possible to provide a proper evaluation of this manuscript in this referee's opinion.

- Thermo-magnetic properties and XRD data for control layers should be included in the manuscript, ie bare LSMO, LCO in the thickness ranges of interest to this paper.

- To illustrate the main claims of this study, it will be important for the authors to report magnetic properties of the P/P interface and compare to P/B interface, if indeed their suggested mechanism is necessary for PMA property. It is not clear to me at present if this is the case or why this particular structure is of relevance.

- How did the authors establish the composition of the LCO layer? at least rough level of materials composition characterization is required to again strengthen their principal claim of the role of the Brownlillerite phase in this paper.

- There appears to be some discrepancy in explanation of figure 4 data set. The authors claim strongest effect appears for a particular thickness, but there is no reason provided for this. I do not understand why this value is special. if this value is indeed special, then how do the authors extract a systematic trend in figure 4b for thickness dependence? Also, how does fig 4d vary for different temperatures? why did the authors choose 10K value to plot this?

- how does data in figure 4 compare with relevance to spintronics applications in the introduction and PMA property sought after in other materials systems? It will be important for the authors to elaborate on their results in the discussion section with connection to the main claims in the introduction.

Responses to the first Reviewer:

Comments: The authors referred to Ref. 31, which reported the magnetic transition of LaCoO_3 at about 75 K, and concluded that the $\text{LaCoO}_{2.5}$ did not contribute the magnetic properties of their heterostructures. It is well known that LaCoO_{3-d} ($\text{LaCoO}_{2.5+d}$) show ferromagnetic transitions up to about 290 K in bulk (S. Balamurugan, et al., PRB 74, 172406, 2006) and 300 K in thin films (N. Ichikawa, et al., Dalton Trans. 41, 10507, 2012). Those previous results indicate that CoO_6 octahedra give ferromagnetic transitions at 290-300 K, which are very close to the magnetic transition temperature of $\text{La}_{2/3}\text{Sr}_{1/3}\text{MnO}_3$ in the present study. Therefore, there still be a possibility that the observed perpendicular magnetic moments of the fabricated superlattices originated from the (partially) oxidized $\text{LaCoO}_{2.5}$ film layer, not the $\text{La}_{2/3}\text{Sr}_{1/3}\text{MnO}_3$ layer. Actually, the oxygen partial pressure during the deposition for $\text{LaCoO}_{2.5}$ in the present study (30 Pa) is quite high compared to that reported in a previous paper.

Responses: (1) Thank the Referee for bringing these two references to our attention. We noticed that these two papers are about SrCoO_x with $2.5 < x < 3$. It is correct that SrCoO_x will show a T_C close to 290 K when x is around 3. However, LaCoO_3 is different from SrCoO_3 . Bulk LaCoO_3 is nonmagnetic at low temperatures and paramagnetic above ~ 110 K (Y. Tokura et al. Phys. Rev. B 58, R1699(1998)). It becomes ferromagnetic only when it was grown on SrTiO_3 or LSAT as a tensile film, showing a Curie temperature of 75 K (D. Fuchs et al. Phys. Rev. B 75, 144402 (2007); Woo Seok Choi et al. Nano Lett. 12, 4966 (2012); Liang Qiao et al. Nano Lett. 15, 4677(2015)). Therefore, if the interfacial $\text{LaCoO}_{2.5}$ was oxidized into the LaCoO_3 phase, it will only have magnetic contributions below 75 K whereas the perpendicular magnetic anisotropy is observed in a wide temperature range below ~ 230 K. Moreover, if the $\text{LaCoO}_{3-\delta}$ phase is formed at interface, the preferred magnetic direction will lie in the film plane rather than being perpendicular to film plane, as will be seen below.

(2) To confirm the unique effect of the perovskite/brownmillerite (P/B) interface on perpendicular magnetic anisotropy (PMA), we fabricated the LCO/LSMO/LCO multilayers with perovskite/perovskite (P/P) interfaces using high preparation oxygen pressures ($P_{\text{O}_2\text{s}}$). We found that when the $\text{LaCoO}_{3-\delta}$ phase prevails, the easy axis rotates from the out-of-plane to the in-plane direction. This result means that the oxidization of the interfacial $\text{LaCoO}_{2.5}$ phase could not produce PMA. On the contrary, the P/P interface favors an in-plane magnetic anisotropy. This conclusion is also supported by the results of density-functional theory calculations (Supplementary Fig. 10f). In the revised manuscript, the result on P/P interfaces was

presented to compare to that of the P/B interfaces.

Fig. 1 (a) A high-angle annular dark field image of the cross sections of the LCO(6nm)/LSMO(6nm)/LCO(6nm) trilayers fabricated under an oxygen pressure of 50 Pa, showing a P/P type interface. Remarkably, dark stripes that indicate structure modulation and the breath mode lattice distortion for Co atoms completely disappear, indicating the absence of LaCoO_{2.5} phase, i.e., the LCO/LSMO interface is of P/P type. (b) Thermomagnetic curves for the LCO/LSMO/LCO trilayers prepared under different P_{O2}s, collected in field-cooling mode with an in-plane (IP) and an out-of-plane (OP) applied field of 0.05 T, respectively. In-plane magnetic anisotropy is observed under the P_{O2} of 50 Pa. (c) Magnetic anisotropy energy as a function of oxygen pressure, showing a monotonic decrease with P_{O2}. The anomaly at P_{O2}=20Pa is due to the degeneration of the magnetism of the LSMO layer when P_{O2} is low.

Comments: The presence of CoO₆ octahedra near the interface should be checked carefully. The observed elongated octahedra might be CoO₆ not MnO₆. The authors are advised to conduct an atomic resolution EELS analysis in the STEM observation and to confirm the exact position of interface (not from the image) and the elongated octahedral is MnO₆. The above discussion is essential for clarifying the origin of the observed perpendicular magnetic moments. Without the confirmation the submitted paper should not be accepted for publication in Nature

Communications.

Responses: Indeed, it is necessary to present the EELS spectra to identify the LSMO interface and to find the location of Mn atoms. We performed an EELS analysis of the atomic distribution across the LCO/LSMO interface and found that the first monolayer just below the interfacial dark stripe is a Mn-O monolayer, though there are slightly interlayer diffusions. According to the suggestion of the Referee, the atomic resolution EELS spectrum images were provided in the revised manuscript (as shown in the Figure below).

Fig. 2 (a) A HAADF image of the LSMO/LCO interface. The interface between the LCO and the LSMO layers is marked by a dashed line. The red arrows denote dark stripes with a $2a_0$ periodicity, which is the typical feature of brownmillerite phase. (b) The corresponding EELS spectrum images of the Mn-L_{2,3} and Co-L_{2,3} edges obtained along the yellow line marked in (a). It is obvious that the octahedra layer just below the LCO/LSMO interface is composed of MnO₆. This result is consistent with the observation of Meyer et al. who found that the first layer connects to the TiO₆ octahedron layer is formed by CoO₄ for the SrCoO_{2.5}/SrTiO₃ interface (Meyer et al. Adv. Electron. Mater. **2**, 1500201 (2016)).

Comments: Figure 2d is difficult to understand, and the authors are requested to add more explanation. Which kinds of lattice parameters are plotted in blue and red in this figure? The extrapolated values (in red?) seem to correspond to the bulk lattice parameters for LSMO (at LCO layer thickness 0) and LCO (at ∞), but the results do not. The obtained $c(\text{LCO}) = 3.77 \text{ \AA}$ does not match either value of A-A distance in Figure 1b. Details explanations are needed.

Responses: The lattice parameters in blue and red in Fig. 2d are respectively in-plane and out-of-plane lattice constants of the SLs. The deduced c_{LCO} is 3.780 \AA , which should be an average of the two A-A distances of the LCO layer shown in Fig. 1b of the original manuscript. The averaged A-A distance is 3.890 \AA , larger than

3.780 Å. Possibly, the brownmillerite phase in the SLs prefers to form in the proximity region of the interface, and the LCO layers are not totally of the brownmillerite structure. As a result, the lattice constant determined by XRD, which provides a macroscopic characterization, is usually smaller than that of the pure brownmillerite phase. The different results of the STEM and XRD have been explained in the revised manuscript. To make room for the EELS spectra and a sketch of the brownmillerite structure, the original Fig. 1b has been moved from Fig. 1 to Supplementary Fig. 1.

Responses to the second Reviewer:

Comments: One of my main comments is why interfacial charge transfer is not considered in this system, whereas it is known to occur in other Mn/Cu interfaces such as YBCO/LCMO (i.e. Chakhalian et al., Science, 318 (2007) 1114-1117). Did the authors investigate it through X-ray Absorption Spectroscopy (XAS) or Electron Energy Loss Spectroscopy (EELS)? What about orbital polarization?

Responses: (1) It is a suggestive comment to consider the effect of charge transfer across the LSMO/LCO interface. We have tried to capture charge transfer by analyzing the valence state of Co based on the technique of EELS, and found that the Co ions in dark stripes are in a slightly lower valence state compared to the ions elsewhere (Please see the Fig. 1 below). However, no considerable signatures for charge transfer are observed at the LSMO/LCO interface. We also performed the XAS analysis and, unfortunately, did not get any reliable results. Probably, the layer thickness of our superlattices is too thick (>6nm) so that the contributions from interfaces are submerged. Despite of these negative results, we still believe that charge transfer has probably occurred. It could a topic of our future work when we optimally design samples. In this manuscript, we presented a short paragraph to discuss the possible effect of charge transfer:

..... Meanwhile, charge transfer may take place between Mn and Co, building up an out-of-plane Co-O-Mn covalent bond³¹ (Charge transfer has been observed across the (Y,Ca)Ba₂Cu₃O₇/La_{2/3}Ca_{1/3}MnO₃ interface, resulting in an orbital reconstruction³⁹). All these will cause a preferred occupation of the d_{3z²-r²} orbital, as confirmed by the DFT calculations (Figure 6b).

(2) We performed the XAS analysis and observed signatures of orbital polarization. However, these data were not presented in the manuscript due to the bad data quality (we only have very limited beam time).

Fig. 1 EELS spectra of the Co ions at dark and bright stripes. The lower peak around 790 eV of the dark stripe Co ions implies a lower valence state.

Comments: It would be useful for the reader to remind the bulk properties of the La(Sr)MnO₃ and LaCoO_{2.5}. What about the magnetic properties of the LaCoO_{2.5} films? In the manuscript it is simply mentioned that no signatures of LCO T_c are observed about the superlattices.

Responses: (1) Thank the Referee for the helpful suggestion. In Fig. 3(a) of the manuscript we have presented the magnetic behaviors of the bare LSMO film (6 nm in thickness). It behaves as bulk LSMO but for a slightly lower Curie temperature due to the thickness effect of the film (326 K versus 350 K).

(2) We failed to get bulk LaCoO_{2.5} phase. LaCoO_{2.5} is unstable. It may appear only in multilayers due to interlayer interaction. In fact, we have tried to prepare the LaCoO_{3- δ} films in a wide oxygen pressure (P_{O₂}) range, from 0.01 Pa to 30 Pa. The main conclusion is that the film is always LaCoO_{3- δ} when P_{O₂} ≥ 10 Pa and no single LCO phase forms when P_{O₂} ≤ 0.1 Pa (Please see the black and red curves in the Figure below. The sample could be a mixture of LCO and other Co oxides). However, the magnetic order of the LCO film is well established when P_{O₂} ≥ 0.1 Pa. Considering the suggestion of the Referee, in the revised manuscript, we have given brief discussions on the LaCoO_{2.5} phase: we would like to emphasize that the LaCoO_{2.5} phase is unstable. It could be induced by interlayer interaction thus exist mainly in the proximity region of the LSMO/LCO interface.

Fig. 2 (a) XRD spectra of the LCO films fabricated under different oxygen pressures. Blue triangles mark unidentified phase. LaCoO₃ phase is obtained above the

oxygen pressure of 10 Pa. (b) Magnetic moments as functions of temperature, measured under an applied field of 0.05 T in the field-cooling mode. Here relatively thick films were prepared to get clear diffraction peaks.

Comments: The authors should also comment on the transport properties of the superlattices investigated. In line 228, the authors mention that their heterostructures' properties contrast with those of metallic films. Are their La(Sr)MnO₃-based superlattices not metallic?

Responses: We are sorry for our misleading presentation. Here the "metallic films" mentioned in our manuscript are the films composed of metals and/or alloys. They are different from our superlattices which are formed by complex oxides. Considering the comments of the Referee, in the present manuscript, we have revised our statements on metallic films: Notably, the PMA in our heterostructure persists significant up to the layer thickness of 10 nm (Figure 4b) in contrast to 2 nm for the films composed of metals or alloys.....

Considering the suggestions of the Referee, we have presented a brief description about the transport properties of the LSMO/LCO superlattices in the discussion section: We have investigated the transport behavior of the SLs. With the decrease of temperature, the resistance of the SLs is well metallic below 300 K like bulk LSMO.

Fig. 3 Temperature dependence of the resistance of the LSMO/LCO superlattices. As expected, the resistance of the SLs is much lower than that of the LSMO counterpart when LCO is ultrathin.

Comments: As indicated by the authors, brownmillerite structures are not as commonly investigated as perovskites. Therefore, I suggest to introduce sketches of the brownmillerite structure as well as the interface in the main figures of the paper.

Responses: This is a very good suggestion. Based on the results of the DFT calculations, a sketch of the brownmillerite structure and brownmillerite/perovskite interface has been presented in Figure 1d of the present manuscript.

Comments:

- Line 26: "relax" instead of "relaxed"
- Line 125: "...LSMO&LCO SIs with a periodicity of 5..." : I guess it should be "number of repetitions" instead of "periodicity"
- Line 142: goal of sketch showing tensile strained is not achieved
- Fig. 3c: explicitly indicate that it corresponds to superlattices (not LSMO)
- Line 228: is it Fig. 4b-citation correct?

Responses: All these mistakes/flaws have been corrected/fixed.

Responses to the third Reviewer:

Comments: - The symmetry mismatch leads to distinct interfacial structures, such phenomena has been discussed in literature for heterophases, e.g Wong et al, J Vac Sci Tech A, 32, 040801, 2014. It will be important for the authors to describe the structure, defects and how the symmetry mismatch is accommodated. This is not clear just looking at the TEM image in figure 1. What kinds of extended defects are formed for instance?

Responses: Thank the Referee for bringing this reference to our attention. It has been inserted into our reference list. Indeed, the interfacial structure should be clearly described to give the reader an idea about how the two different structures accommodate each other. In Fig. 1d we presented a sketch on the LSMO/LCO interface. Furthermore, in supplementary Fig. 2 (also shown in the Figure below), we have presented a layer by layer graphic illustration to show how the CoO_4 layer connects to the MnO_6 layer, and what will be resulted by their symmetry mismatch. Meanwhile, we have also provided two short paragraphs in the main text of the revised manuscript to explain thus resulted lattice distortions:

It will be interesting to see how the CoO_4 tetrahedra and the MnO_6 octahedra accommodate each other at the LCO/LSMO interface. As shown in Supplementary Figure 3, just below the interface is a MnO_6 layer; each MnO_6 links to four neighboring MnO_6 octahedra through its four corner oxygen atoms. On the other side of the interface, CoO_4 tetrahedra form another layer in parallel to that of MnO_6 . However, each CoO_4 is only connected to two neighbors, yielding a breath mode lattice distortion in the CoO_4 network as demonstrated by the HAADF image in Fig. 1a.

.....Notably, a CoO_4 in one layer has an exclusive corresponding MnO_6 in adjacent layer, connecting the latter through the apical oxygen atom (Supplementary Figure 2). Since there is unfilled space in the CoO_4 layer, the neighboring MnO_6 will adjust its orientation and shape to minimize elastic energy. A result of this is the tiling around [110] axis and an accompanied elongation of the MnO_6 (Fig. 1d). In general, the elongation particularly the octahedron tilting will cause a chain reaction, reversely titling the next MnO_6 layer. This means that interface effect is not limited to interfacial layer, extending to considerably distant layers though it will decay with the distance from interface.....

Fig. 1 A sketch showing how the MnO_6 and CoO_4 layers join one another based on the results of DFT calculations. (a) A single MnO_6 layer. (b) A single CoO_4 layer. (c) The CoO_4 layer is placed above the MnO_6 layer. Here the La/Sr atom has been dropped for clarity. As indicated in (b), the nearest Co-Co distance is shorter in the CoO_4 chain and longer between two neighboring chains. This causes a regular misalignment of top Co and bottom Mn atoms shown in (c). For example, the first row of Co atoms exhibits a bottom left shift while the second row displays an up left shift. Consequently, the MnO_6 octahedron has to tilt around the $[1-10]/[110]$ axis (marked by a red arrow in (a)) and elongates along the $[001]$ axis.

Comments: There is no discussion on how inter-diffusion can affect the properties reported here. For instance, the authors claim there is no effect for the first few layers, and then the anisotropy effect kicks in. It will be important to explain to the reader why this is the case. Is this related to strain, chemical in-homogeneity, unknown phase formation at interface etc.

Responses: According to our experiments, PMA is absent when the LCO layer is ultrathin and kicks in when LCO is thicker than ~ 1.5 nm. It is possible that the brownmillerite phase is not formed when LCO is too thin thus the PMA is not established. This is understandable since the c-axis lattice constant of the brownmillerite unit cell is large, ~ 1.5 nm. This explanation is supported by the observation of the LCO/LSMO superlattice with a LCO layer of 4uc.

Fig. 2 (a) EDX mapping of the superlattices shows the presence of LCO layers. However, from the HAADF image (b) of the corresponding superlattices we observed no signatures of brownmillerite phase when LCO layer is 4 uc in the LSMO/LCO superlattices.

Indeed, information on interlayer diffusion is important. We performed an EELS analysis of the LCO/LSMO interface, as shown in the Figure below, and found a weak Mn tail in the nearest neighboring LCO unit cell of the interface. In general, a so minor interlayer diffusion is unable to strongly affect interfacial distortions because of the similar ionic radii of Mn and Co (0.645 Å for Mn³⁺ and 0.61 Å for Co³⁺ in high spin state) thus destroys the PMA.

With the EELS data being presented, we have made minor revisions to our manuscript: We examined the EELS spectrum of the multilayers and believed that the weak interlayer diffusion is unable to destroy the effect of LCO on LSMO. Possibly, the brownmillerite structure is not well established when LCO is 0.8 nm since the c-axis lattice constant is ~1.5 nm for a brownmillerite unit cell.

Fig. 3 (a) Lattice image around the LCO/LSMO interface. (b) EELS spectrum images of the Mn-L_{2,3} and Co-L_{2,3} edges, corresponding to the yellow line area of the HAADF image shown in (a).

Comments: Thermo-magnetic plots are shown for various fields and T in the main and supplement. It will be useful for the authors to analyse this and present a summary. For instance, is the T_c changing with interfacial layer structure or number of layers? In some cases, I see the curves crossing one another, but this is not explained.

Responses: This is a helpful suggestion. We have summarized the Curie temperature of differently structured multilayers and presented the results thus obtained in Supplementary Fig. 9. Correspondingly, in the revised manuscript we have given a comment to the variation of the Curie temperature: We noted that in Figs. 4a and 4b the Curie temperature of the trilayers monotonically grows with t_{LSMO} and keeps nearly constant as t_{LCO} increases (Supplementary Figure 9). This is understandable since the Curie temperature of the trilayers is mainly determined by the LSMO layer.

Indeed, the in-plane and out-of-plane M-T curves sometimes cross one another. This is due to the different influences of spin reorientation on them. When just cooled below the Curie temperature, the magnetic moment increases more rapidly along the in-plane direction than along the out-of-plane direction. However, as the spin reorientation temperature is approached, the out-of-plane magnetic moment grows drastically whereas the in-plane one drops. Consequently, the two curves in these two directions cross one another at a specific temperature. In revised manuscript, we have given a brief explanation to this phenomenon in the caption of Fig. 3: Two M-T curves recorded along the in-plane and the out-of-plane directions, respectively, sometimes cross one another since the spin reorientation has changed relative variations of the magnetic moment with temperature along

these two directions.

Fig. 4 Temperature dependences of magnetic moments of the LSMO/LCO trilayers, obtained in field-cooling mode with an in-plane and an out-of-plane applied field of 0.05 T, respectively. Crossing one another of these two curves is due to spin reorientation. The Minor step at ~75 K indicates the presence of residual $\text{LaCoO}_{3-\delta}$ phase.

Comments: It will be helpful for the authors to elaborate on the principal result. It is not clear to me why this result is particularly significant or some comparison to existing spintronic technologies or how the anisotropy constant compares to what is useful or impressive in this class of materials or others etc? Without this information properly discussed in the context of the introduction and claims, it is not possible to provide a proper evaluation of this manuscript in this referee's opinion.

Responses: This is a valuable suggestion. In the revised manuscript, the significance of the present work has been explained in the section of introduction: For the first time, this work demonstrated the great potential of symmetry engineering in the exploration for novel effects in magnetic complex oxides. Its application to, as an example, the LSMO/LCO multilayers has produced a strong perpendicular magnetic anisotropy (PMA), which has been in hot pursuit of spintronics, of the LSMO layer which is otherwise of easy plane; the maximal PMA energy is $\sim 1.3 \text{ J/cm}^3$, which is more than one order of magnitude higher than that achieved via the conventional approaches such as magnetoelastic coupling (from 0.01 to 0.1 J/cm^3)³²⁻³⁵ and magnetocrystalline anisotropy ($\sim 0.018 \text{ J/cm}^3$)³². This large PMA stems from the symmetry mismatch of the MnO_6 and CoO_4 layers at the LSMO/LCO interface, which results in cooperative distortions of the interfacial

oxygen polyhedra as evidenced by high resolution lattice structure analysis and density-functional theory (DFT) calculations. Moreover, the symmetry break at the P/B interface is also expected to bring about emergent phenomena associated with directional electronic/ionic transport, two-dimensional electric polarization and two-dimensional magnetism and etc., thus opens a new avenue for the exploration for new concepts of physics and materials.

Comments: Thermo-magnetic properties and XRD data for control layers should be included in the manuscript, i.e., bare LSMO, LCO in the thickness ranges of interest to this paper.

Responses: This is a good suggestion. In the revised manuscript, we have presented the M-T curves and the XRD data in Supplementary Fig. 5 for the typical LCO and LSMO bare films. Meanwhile, the M-T curves under different fields were also given in the revised manuscript for a bare LSMO film of 6nm (Fig. 3a)

Comments: To illustrate the main claims of this study, it will be important for the authors to report magnetic properties of the P/P interface and compare to P/B interface, if indeed their suggested mechanism is necessary for PMA property. It is not clear to me at present if this is the case or why this particular structure is of relevance.

Response: This is a very good suggestion. We prepared the LCO/LSMO/LCO trilayers under the oxygen pressures ranging from $P_{O_2}=20$ Pa to 50 Pa. Here high oxygen pressures up to 50 Pa were used to get perovskite LCO phase. Indeed, we observed a monotonic decrease of the PMA energy with the increase of oxygen pressure, and the in-plane anisotropy prevails under the P_{O_2} of 50 Pa. The STEM analysis shows that we obtained the P/P type multilayers when P_{O_2} is 50 Pa. The typical results have been presented in revised manuscript (Fig. 5).

Comments: How did the authors establish the composition of the LCO layer? at least rough level of materials composition characterization is required to again strengthen their principal claim of the role of the Brownlillerite phase in this paper.

Responses: It is a good suggestion to prove that the LCO is the brownmillerite phase by determining its composition. For the bare LCO film, the La to Co ratio is close to 1:0.96 as determined by EDX analysis. We believed that the La to Co ratio in multilayers is similar as that of the bare film. However, it is much more challenging

to determine the oxygen content in the LCO layer. As an alternative, we analyzed the valence state of the Co ions of the LCO(6nm)/LSMO(6nm)/LCO(6nm) trilayers by XPS and found obvious signals of Co^{2+} , as indicated by the appearance of shake up satellite peaks (marked by arrows). This could be the indirect evidence for the appearance of brownmillerite phase. This result has been included in Supplementary Figure 1c of the revised manuscript.

Fig. 5 Spectra of the X-ray photoelectron spectroscopy (XPS) of the LCO/LSMO/LCO trilayers and the LCO bare layer (6nm). Arrows mark the characteristic shake up peaks of Co^{2+} .

Comments: There appears to be some discrepancy in explanation of figure 4 data set. The authors claim strongest effect appears for a particular thickness, but there is no reason provided for this. I do not understand why this value is special. If this value is indeed special, then how do the authors extract a systematic trend in figure 4b for thickness dependence? Also, how does fig 4d vary for different temperatures? Why did the authors choose 10K value to plot this?

Responses: We are sorry for the ambiguous presentation of the manuscript. Fig. 4b shows the variation of the PMA constant with t_{LSMO} when t_{LCO} is fixed to 6 nm. Fig. 4d shows the variation of PMA constant with t_{LCO} when t_{LSMO} is fixed to 6 nm. To avoid misleading the readers, we have replaced the original statement "The strongest effect occurs when $t_{\text{LCO}}=6$ nm,....." by "The strongest effect occurs when $t_{\text{LCO}}=6$ nm for this series of trilayers....." .

The thickness of 6 nm for the LCO is not special. Here we simply want to show a fact that a too thin or a too thick LCO layer in the multilayers disfavors the PMA. To avoid misleading, a statement has been presented "This result shows that a too thin or a too thick LCO layer in the multilayers disfavors the PMA."

At temperatures well below the Curie temperature, the PMA is nearly temperature independent. We chose the data of 10 K simply because most of our M-H curves were recorded at this temperature. At this temperature, our measuring system is most easily thermally stabilized. This has been explained in the Figure captions of Figs. 3 and 4 of the revised manuscript.

The general $K_{\text{tot}}-t_{\text{LCO}}$ relation in Fig. 4d is not affected by temperature if the temperature is far below the Curie temperature though the detailed dependence is slightly different, as shown in the following Figures.

Fig. 6 Magnetic anisotropy energy as a function of the layer thickness of LCO, collected at three typical temperatures of 10, 50, and 100 K.

Comments: how does data in figure 4 compare with relevance to spintronics applications in the introduction and PMA property sought after in other materials systems? It will be important for the authors to elaborate on their results in the discussion section with connection to the main claims in the introduction.

Responses: Thank the suggestive comments. In the revised manuscript, a short paragraph has been added to in the discussion section to echo the claims in the introduction:

As an example, a natural extension of the LSMO/LCO superlattices may be perpendicular magnetic tunnel junctions where the LSMO layers serve as electrodes with a perpendicular spin orientation and the LCO layer acts as a barrier layer that is highly insulating at low temperatures. For this kind of magnetic junction, it will be much easier for electrical current to reverse the spin direction of the LSMO layer, yielding electric field-controllable magnetoresistance (We have investigated the transport behavior of the SLs. With the decrease of temperature, the resistance of the SLs is well metallic below 300 K like bulk LSMO. In contrast, LCO is highly semiconducting below 300 K). An advantage of the present SLs over

the typical metallic film for perpendicular magnetic tunnel junctions is that its PMA has a large tolerance of film thickness, occurring in the thickness range from 3 to 10 nm, rather than being limited to ~2 nm. Moreover, noting the unique characters of the brownmillerite oxides as ionic conductors, catalyzers, and oxygen separation membranes, amazing effects in addition to perpendicular magnetic anisotropy are expected for the P/B combinations. For example, the symmetry break at the P/B interface may bring about emergent phenomena associated with directional electronic/ionic transport, two-dimensional electric polarization and two-dimensional magnetism and etc., thus opens a new avenue for the exploration for new concepts of physics and materials.

Reviewers' Comments:

Reviewer #1:

Remarks to the Author:

J. Zhang, et al. have revised the manuscript according to comments by the reviewers.

The first reviewer confused the results of LaCoO₃ and SrCoO₃ in the previous comments, and the authors now made the confusion clear. Because the La_{2/3}Sr_{1/3}MnO₃ contains Sr at the A site, there is still a possibility of SrCoO₃-type structure at the interface. However, the additional experimental result on multilayers prepared under high oxygen pressure convinced the reviewer of the authors' conclusion.

In response to the reviewer's comments, the authors have also provided additional EELS experimental results and explanations for the figures.

Therefore, the revised manuscript appears to be now suitable for publication in Nature Communications. The reported results will give important information to researchers in the related fields.

Reviewer #2:

Remarks to the Author:

In their detailed reply, the authors addressed all my comments and suggestions. Therefore, I recommend now the publication of the paper.

Reviewer #3:

Remarks to the Author:

The authors have addressed my comments from the first review. The persistent weakness of the manuscript is lack of evidence to show the brownmillerite phase, which is presently indirectly evidenced from Co²⁺ satellite peaks. It will be beneficial if the authors could provide stronger evidence for this phase. If that is not possible, at least present XPS data for the perovskite Co containing phase to prove the satellite peaks only appear for the B phase.

Responses to the third Reviewer:

Comments: The authors have addressed my comments from the first review. The persistent weakness of the manuscript is lack of evidence to show the brownmillerite phase, which is presently indirectly evidenced from Co^{2+} satellite peaks. It will be beneficial if the authors could provide stronger evidence for this phase. If that is not possible, at least present XPS data for the perovskite Co containing phase to prove the satellite peaks only appear for the B phase.

Responses: Following the suggestion of the Reviewer, we collected the XPS spectrum of the LCO/LSMO/LCO trilayers prepared under a high oxygen pressure of 50 Pa. The LCO layer in this sample is perovskite-structured as evidenced by the STEM analysis (Fig. 5a of the manuscript). As shown in the following Fig. 1, no signatures of shake up satellite peaks are observed (blue curve). This is different from the LCO/LSMO/LCO trilayers with modulated lattice structures (red curve). For comparison, the XPS spectrum of perovskite LaCoO_3 is also presented (black curve). For this sample there are no satellite peaks either. These results show that the satellite peaks do not appear in perovskite LaCoO_3 phase.

Considering the comments of the Reviewer, we added the following statements to the revised manuscript: As stated above, the typical shake up satellite peaks of the Co^{2+} ions have been observed in the XPS spectrum of the trilayers with modulated lattice structures, which are the lateral evidence of the B-typed structure of the LCO layer. In contrast, no satellite peaks are observed for the LCO/LSMO/LCO trilayers prepared under 50 Pa (Supplementary Figure 1c). This is consistent with the result of STEM analysis, which suggests that the LCO layer in this sample is perovskite-structured.

Accordingly, the XPS spectrum of the LCO/LSMO/LCO trilayers prepared under 50 Pa was added to Supplementary Figure 1c.

Fig. 1 Spectra of X-ray photoelectron spectroscopy (XPS) of the Co ions in the LCO(6nm)/LSMO(6nm)/LCO(6nm) trilayers prepared under the oxygen pressures of 30 Pa and 50 Pa, respectively. Obvious shake up satellite peaks (marked by two red triangles) are observed for the former trilayers but not for the latter trilayers. This result confirms the presence of considerable Co^{2+} in the former sample, for which LCO is expected to be brownmillerite-structured.

Reviewers' Comments:

Reviewer #3:

Remarks to the Author:

The authors have included the requested XPS data to show the presence of satellite peaks in select samples. I do not have any other comments.